# Public Health Midwives' perspectives on paediatric snakebite prevention and management in rural Sri Lanka: A qualitative study

Kavinda Dayasiri[1]*, Achila Ranasinghe[1], Tharuka Perera[1], Indika Gawarammana[2], Shaluka Jayamanne[1]

**1** Faculty of Medicine, University of Kelaniya, Ragama, Sri Lanka, **2** Faculty of Medicine, University of Peradeniya, Peradeniya, Sri Lanka

* kavindadayasiri@gmail.com

## Abstract

### Background

Snake envenoming remains a significant yet under-addressed public health issue in rural Sri Lanka, where children face disproportionately severe outcomes. Public Health Midwives (PHMs) serve as trusted community health workers. While they may be well placed to support paediatric snakebite prevention and early response, their actual capacity and role in this regard remain underexplored.

### Methodology/Principal findings

This qualitative study explored PHMs' insights into paediatric snakebite prevention and management in two high-incidence districts—Ampara and Polonnaruwa. A total of 74 PHMs participated in 11 focus group discussions (FGDs) guided by a semi-structured thematic framework. Thematic analysis revealed five major themes. PHMs reported limited awareness of paediatric-specific snakebite risks and described frequent community reliance on traditional beliefs that delayed care. Their first-aid knowledge was often outdated or incomplete, and they lacked practical training. Operational challenges—such as excessive workload, language barriers, and poor infrastructure—limited outreach. Despite these constraints, PHMs expressed strong motivation to engage in prevention through schools and home visits. They proposed feasible solutions including targeted training, improved transport and referral systems, and culturally appropriate health education materials, highlighting their readiness to contribute if supported with system-level interventions.

### Conclusions/Significance

Public Health Midwives in rural Sri Lanka have potential to reduce paediatric snakebite harm but face key systemic barriers. Effective prevention requires targeted

**Data availability statement:** All de-identified data provided within the research report and Supporting information.

**Funding:** The author(s) received no specific funding for this work.

**Competing interests:** The authors have declared that no competing interests exist.

training, supportive tools, and integration into existing services without adding workload. Strengthening PHM capacity through brief educational interventions within primary care systems offers a practical, sustainable approach to improve outcomes in these communities.

## Author summary

Snake envenoming is a neglected tropical disease that disproportionately affects rural populations in South Asia, with children facing higher morbidity and mortality. In Sri Lanka, Public Health Midwives serve as vital community-level health workers with potential to influence prevention and first-aid responses. This qualitative study explored PHMs' perspectives on paediatric snakebite management in the high-incidence districts of rural Sri Lanka. Eleven focus group discussions were conducted with 74 PHMs using a semi-structured guide. Thematic analysis revealed five key themes: limited awareness of paediatric-specific snakebite risks, widespread reliance on traditional and spiritual remedies, inadequate first-aid training and confidence, logistical and systemic challenges to community engagement, and actionable recommendations for improving care. PHMs described frequent delays in treatment due to community myths and inadequate rural health infrastructure, but expressed willingness to take on greater roles in education and advocacy. They proposed feasible solutions, including school-based awareness, strengthened emergency transport, hospital preparedness, and integration of snakebite education into routine PHM activities. The findings highlight the need to empower PHMs through structured training, culturally appropriate communication tools, and systemic support. Doing so offers a sustainable and community-driven path to reducing the burden of paediatric snake envenoming in Sri Lanka and similar settings across the global snakebite belt.

## Introduction

Snake envenoming is a neglected tropical disease that disproportionately affects rural and impoverished populations across the globe, causing over 5 million bites and up to 138,000 deaths annually, with many more left with permanent disabilities [1]. Children, typically defined as individuals aged 0–18 years, are particularly vulnerable due to their smaller body mass, greater outdoor exposure, and limited ability to respond to danger [2]. Paediatric snakebites represent a substantial proportion of cases in high-incidence regions, and their clinical impact is significant, often requiring specialised assessment and timely management [3]. Despite these risks, paediatric-specific data on snakebite incidence, outcomes, and prevention strategies remain limited in most settings.

In Sri Lanka, a nationwide community survey estimated around 80,500 annual snakebites, including 30,500 envenomings and approximately 460 deaths, though

age-specific data are scarce [4]. Spatial risk mapping highlighted considerable geographic variation in incidence, underscoring the need for locally tailored responses [4]. Despite this, paediatric-specific data and preventive strategies remain scarce within the country's broader snakebite response framework [5].

Public Health Midwives are frontline health workers in Sri Lanka's public health system, primarily responsible for providing maternal and child health services to pregnant women and children under five years within their communities. While disease surveillance is typically the role of Public Health Inspectors, PHMs' close contact with families positions them as potential agents for health education and early intervention. However, their preparedness, motivation, and capacity to engage in paediatric snakebite prevention and management have not been thoroughly explored.

Building on earlier components of this research programme—which highlighted poor parental knowledge of snakebite first aid, reliance on traditional healing practices, and delays in seeking formal care for affected children in rural Sri Lanka [6–8]—this study turns attention to a critical but understudied cadre within the public health system: Public Health Midwives. While previous findings also revealed gaps in clinical preparedness and systemic challenges in responding to paediatric envenoming, little is known about how PHMs perceive their role in prevention or early response, particularly beyond their formal mandate of supporting maternal and child health services for children under five [9,10]. Given their close engagement with families and established trust within communities, PHMs may represent a valuable yet underutilized resource in reducing paediatric snakebite harm. This study explores their knowledge, experiences, and perceived barriers in addressing paediatric snakebites, and captures their recommendations for improving prevention and early care strategies within their operational realities.

## Methods

### Ethics statement

Ethical approval for the study was obtained from the Ethics Review Committee of the Postgraduate Institute of Medicine, University of Colombo (Reference: ERC/PGIM/2024/080). We have adhered to strict ethical principles throughout this project, including maintaining the confidentiality of fieldwork locations, which will not be named. All identifiable information about individual participants—such as names, contact numbers, and places of residence—was either removed or pseudonymised in all datasets. Each participant was assigned a unique Participant ID. All participants were informed of the study's purpose, confidentiality measures, and their voluntary right to withdraw at any time. Written informed consent was obtained from all PHMs prior to participation. No financial incentives were provided; however, refreshments were offered at the conclusion of discussions. Patients and/or the public were not involved in the design, or conduct, or reporting, or dissemination plans of this research.

### Study design

The study employed a qualitative exploratory design using focus group discussions (FGDs) to investigate the perspectives of community health workers—specifically Public Health Midwives—on the prevention, first aid, and management of paediatric snakebites in rural Sri Lanka. PHMs were selected given their unique household-level access to families, especially those with young children, and their continuous engagement in maternal and child health services within the community. The aim was to elicit nuanced insights into PHMs' awareness, knowledge gaps, engagement challenges, traditional belief encounters, and recommendations for strengthening community-level responses to snake envenoming in children.

### Participant recruitment and selection

The study was conducted in the Ampara and Polonnaruwa districts, two rural regions in Sri Lanka identified through national epidemiological surveillance as high-risk areas for venomous snakebites (Fig 1) [4]. These districts have a predominantly agrarian economy, with human-snake encounters commonly occurring near paddy fields, domestic

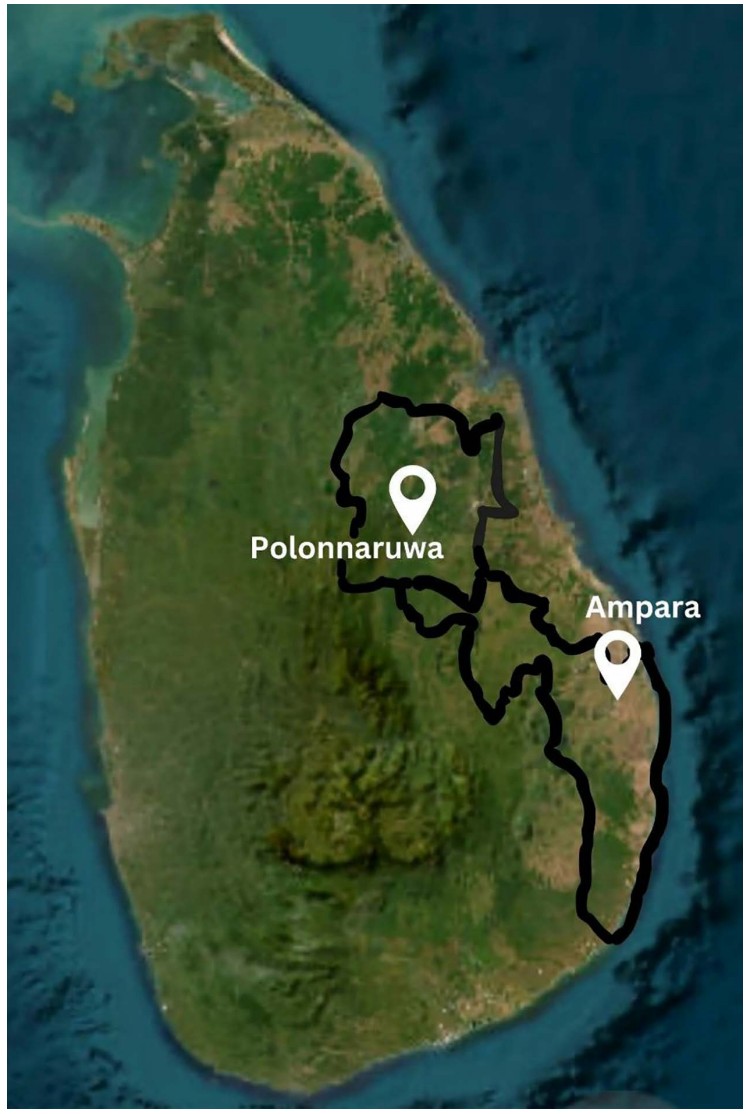

**Fig 1. Distribution of the study settings in Sri Lanka.** Map base layer from the U.S. Geological Survey (Public Domain), available at: https://earthexplorer.usgs.gov/.

gardens, and in poorly sealed homes. Public Health Midwives in these areas serve as key frontline health workers, responsible for maternal and child health surveillance, health education, and community outreach across dispersed rural populations.

A total of 74 PHMs were purposively selected to participate in eleven focus group discussions (FGDs), with each group comprising 5–7 participants. All participants were currently employed by the regional health services and had a minimum of one year of field experience in their respective communities. Selection criteria emphasized geographical diversity across snakebite hotspots and representation from both Sinhala and Tamil-speaking communities. Participants were identified and invited through the support of regional Medical Officers of Health (MOHs). There were no refusals or withdrawals. All PHMs had prior experience conducting home visits, providing first-aid advice, and interfacing with families affected by or at risk of snakebites.

### Facilitator characteristics and reflexivity

Focused group discussions were facilitated by the principal investigator—a male physician with a clinical and academic background in paediatric toxicology and qualitative research—alongside a trained public health assistant with experience in rural fieldwork. Both facilitators were fluent in Sinhala and worked with a Tamil-speaking interpreter when necessary. The facilitators had no prior supervisory or evaluative relationship with any PHMs, helping to ensure open, non-hierarchical dialogue. Reflexive field notes were maintained to account for positionality and group dynamics.

### Data collection procedures

All focused group discussions were held in neutral, community-based locations familiar to the participants, including MOH offices and health education centres. Each session lasted approximately 45–60 minutes. A semi-structured interview guide was used to explore six key domains: (1) existing community-level preventive practices; (2) traditional beliefs and care-seeking behaviour; (3) PHMs' awareness and first-aid knowledge; (4) perceptions of healthcare accessibility and emergency response systems; (5) perceived challenges in community engagement; and (6) recommendations for improving snakebite management in children. The discussion guide was pilot-tested with three non-participating PHMs from a neighbouring district to ensure cultural appropriateness and clarity. All discussions were audio-recorded with consent and supplemented by detailed field notes documenting non-verbal cues and group dynamics.

### Data analysis

All audio recordings were transcribed verbatim in the original language and subsequently translated into English for analysis. Thematic analysis followed Braun and Clarke's six-step framework [11]. Initial familiarization was followed by open coding of transcripts to identify relevant textual units. Codes were iteratively reviewed and grouped into broader themes and subthemes, with particular attention to recurrent patterns, divergent views, and context-specific insights. Two investigators who facilitated the FGDs also conducted the initial coding and thematic analysis. To enhance rigor and minimize bias, coding was performed independently by both researchers, with discrepancies resolved through discussion. Thematic saturation was considered achieved after the ninth focused group discussion, as no new themes emerged in the final two groups, which were used to validate thematic consistency.

### Validation and trustworthiness

To enhance credibility, preliminary thematic summaries were shared with a subset of participating PHMs (n = 6). These member-checking sessions allowed participants to confirm or refine the accuracy of the interpretations. All respondents endorsed the thematic findings and offered minor clarifications, which were integrated into the final analysis. Reflexivity, triangulation of analysts, and use of direct quotations helped enhance analytic trustworthiness.

## Results

### Participant characteristics

A total of 74 Public Health Midwives participated in the focus group discussions. The mean age of participants was 39.8 years (SD = 8.6). In terms of work experience, 32.4% (n = 24) had less than five years of service, 28.4% (n = 21) had between 5 and 10 years, 24.3% (n = 18) had between 10 and 15 years, and 14.9% (n = 11) had more than 15 years of experience. Less than half of the participants (41.9%, n = 31) reported having previously provided snakebite-related health education to parents. Only a small number (6.8%, n = 5) had received prior training specifically on first aid for paediatric snakebites. Participants were drawn from two high-risk districts: 47.3% (n = 35) from Polonnaruwa and 52.7% (n = 39) from Ampara.

The inductive thematic analysis of the 11 focus group discussions revealed five major themes (Fig 2).

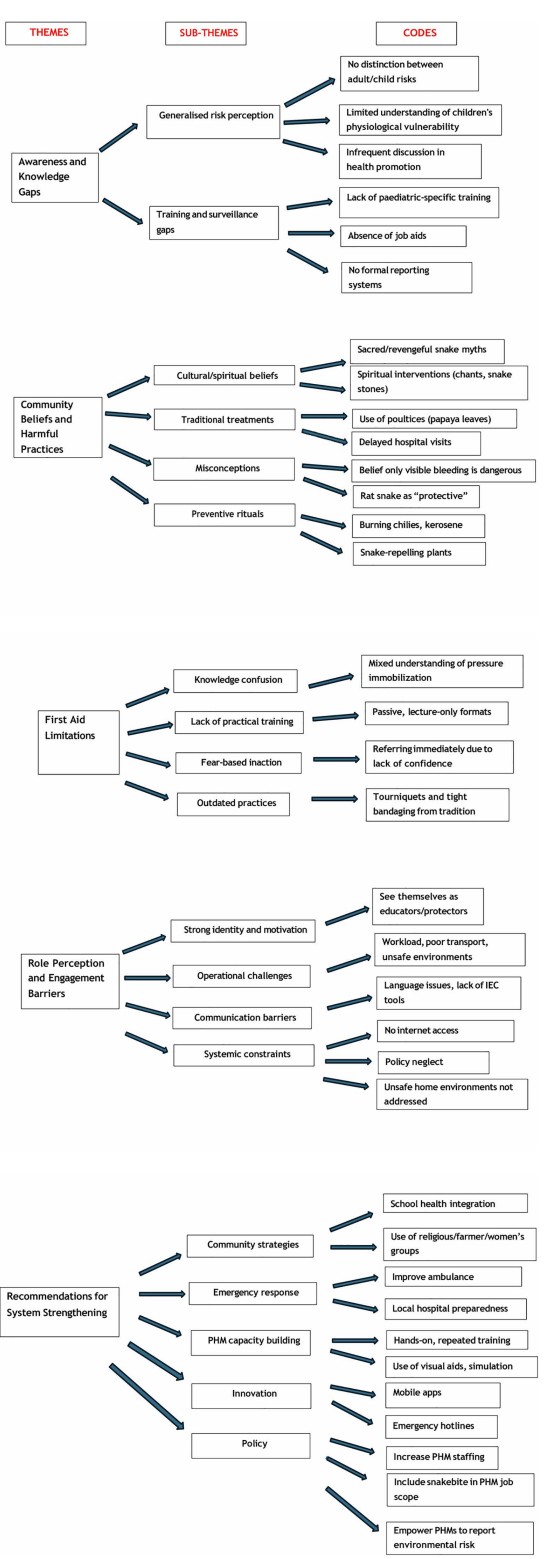

**Fig 2. Thematic map of Public Health Midwives' perspectives on paediatric snakebite prevention and management.**

1.  **Awareness and knowledge gaps regarding paediatric snakebite risks.** Most midwives described snakebite risks in general terms and did not distinguish between adult and child exposures. "*We know snakebites happen here, but we are not told why children are more affected. We just look after them the same way.*" (PHM, 34) Only a few PHMs had received formal training addressing the differences in signs, symptoms, and clinical progression of envenoming between children and adults. Most relied on anecdotal knowledge gained through community conversations or personal experience. There was limited familiarity with medically important snake species in their region, and virtually none had access to structured or updated guidance on identifying venomous versus non-venomous bites. "*Many parents lack knowledge about identifying snakes, and many believe that a snakebite is only serious if there is bleeding. However, I have seen a case in my area where someone was bitten by a snake at night. He and his family didn't act quickly, and by the time they brought him to the hospital, he was already dead.*" (PHM, 65)

Surveillance and reporting systems for paediatric snakebite incidents were also inadequate. PHMs indicated they often became aware of cases only informally—through word of mouth or after the child was already hospitalized—rather than through any structured referral or notification mechanism. "*We hear from the parents that a child got bitten, but there is no system for us to track or report it.*" (PHM, 15) The knowledge gaps extended beyond clinical recognition to the prevention of exposure. For instance, although midwives knew that snakes tend to enter homes during certain seasons, few could articulate specific strategies that would protect children in particular—such as using raised beds, ensuring secure flooring, or encouraging protective footwear for outdoor play. Many participants attributed these deficits to the absence of paediatric-focused content in general health education materials and a lack of targeted training sessions for frontline workers. As a result, snakebite prevention and first aid were rarely emphasized in child health promotion work, despite the obvious risk.

2.  **Community-level beliefs and harmful practices.** Traditional beliefs surrounding snakes and snakebites remain deeply entrenched in rural Sri Lankan communities, strongly influencing household behaviours and healthcare-seeking responses—particularly in paediatric cases. Public Health Midwives consistently highlighted the tension between biomedical knowledge and longstanding local customs, which often led to harmful delays in accessing appropriate medical care for snakebite victims. A significant number of community members, especially elders, continued to favour indigenous healing methods over prompt hospital treatment. These included visiting a traditional healer who might chant incantations, place a "snake stone" on the bite site, administer herbal remedies, or perform spiritual rituals. "*They believe the stone can pull out the venom. Even if we say go to the hospital, they waste time doing these things first.*" (PHM, 39) One particularly concerning belief reported by PHMs was that assaulting the person who carries the patient to the healer could somehow reduce the effect of venom—a notion rooted in spiritual symbolism rather than biology. "*Some believe that assaulting the message carrier to the traditional healer will reduce the toxin inside the body of the patient.*" (PHM, 13) Numerous myths were linked to the spiritual significance of snakes, especially cobras and kraits. For instance, it was commonly believed that harming certain species could bring bad luck or provoke revenge from other snakes. "*They think if you kill a cat snake, seven more will come. So, they are afraid even to chase it away.*" (PHM, 16) "*Some believe the cobra is sacred and will protect treasure. If you kill it, it brings misfortune.*" (PHM, 5)

Another unusual belief was that rat snake bites immunise individuals against other venomous snakes, leading some community members to downplay or ignore bites from supposedly "harmless" species. "*Some villagers believe if a rat snake bites you, no other venomous snake will bite you again. So, they don't take it seriously.*" (PHM, 61) Such beliefs often resulted in significant treatment delays, as families experimented with home remedies or sought spiritual interventions before considering medical care. In paediatric cases, these delays proved particularly dangerous. PHMs described instances where herbal poultices, such as papaya leaves, were applied and tightly bandaged over the bite—worsening local tissue damage or allowing systemic envenoming to progress. "*In one case, they put papaya leaves and wrapped it tightly, and only came to hospital after the child started shaking.*" (PHM, 44)

While younger parents were reportedly more open to biomedical guidance, older relatives often retained decision-making authority, especially in multigenerational households. PHMs expressed that directly contradicting elders or

dismissing their practices often provoked resistance and disengagement. "*If we say 'this is wrong,' they stop listening. We have to find a way to respect their beliefs while slowly changing them.*" (PHM, 19) The midwives also noted that certain environmental and religious practices were perceived as protective. These included burning chilies, spreading kerosene oil around houses, or growing specific plants believed to repel snakes—such as marigolds, long coriander, and Devil's trees.

"*Some grow Marigold flowers, long coriander and Devil's trees. They believe it keeps snakes away.*" (PHM, 27) "*During the rainy season, they clean around the house and apply kerosene oil in front of doors and windows to chase away snakes.*" (PHM, 53) While some of these practices may contribute indirectly to environmental hygiene and deterrence, many are based on superstitions that do not prevent or mitigate envenomation. Nevertheless, they are often deeply valued and passed down through generations.

PHMs described how traditional beliefs, family hierarchies, and community norms strongly influenced healthcare-seeking behaviours following paediatric snakebites. They felt that efforts to improve outcomes must be rooted in cultural understanding and shaped by the local context. According to the PHMs, dismissing these beliefs outright could alienate families and reduce the effectiveness of interventions. Instead, they stressed the importance of introducing counter-narratives that are respectful of tradition while promoting evidence-based care, ideally through collaboration with trusted community leaders.

**3. First aid knowledge and training limitations.** Although nearly all PHMs had heard of basic first aid principles for snakebites—such as washing the bite site and calming the patient—there was considerable variation in actual knowledge, confidence, and application. Many expressed uncertainties about the correct sequence of actions or whether certain practices, such as applying pressure bandages, were recommended for children.

"*I know we should wash it, but some say you must tie above the bite. Others say never tie. So, we are confused.*" (PHM, 47) Only a few participants mentioned immobilising the limb or advising patients to stay still—key components of WHO-recommended first aid. None of the PHMs mentioned having received hands-on training with paediatric mannequins or real-life simulations. Several had attended lectures or Zoom sessions, which they found insufficient for skill-building. "*They gave us a series of lectures, but we need to practice. I have never applied a bandage on a child after a bite.*" (PHM, 58) A recurrent concern was the fear of doing harm. This led many PHMs to default to advising immediate transport to the hospital without attempting any first aid themselves. "*I tell them go to hospital quickly. I don't want to make it worse by doing something wrong.*" (PHM, 7) Some reported using outdated or potentially harmful practices due to a lack of clarity. For example, a few still advised tight bandages based on community norms. "*We were told years ago to tie tightly. Now some say that's wrong, but we're not sure.*" (PHM, 42) The lack of formal and consistent training was cited as a key barrier, and most PHMs expressed a strong desire for updated, child-focused, and practical training sessions to build their competence and confidence.

**4. Role perception and practical challenges in community engagement.** Public health midwives consistently viewed themselves as integral, trusted actors in their communities—well-positioned to influence household behaviours and promote child safety, including in relation to snakebite prevention and first aid. However, their ability to fulfill this role was constrained by a range of practical, systemic, and contextual challenges that undermined their engagement at the community level. Foremost among these was workload pressure and competing priorities. PHMs are already tasked with an extensive portfolio of responsibilities, including maternal and child health surveillance, immunisation programs, nutrition counselling, and chronic disease monitoring. Adding snakebite prevention without dedicated time, resources, or formal policy backing was seen as unrealistic and burdensome by some. "*We already have so much to do—pregnant mothers, children, home visits, clinics. Where can we fit snakebite education?*" (PHM, 20)

Geographic and environmental constraints further compounded their difficulties, especially in remote or underserved areas. Poor road infrastructure, scattered settlements, and seasonal barriers—such as flooding or roaming elephants—often limited midwives' ability to conduct home visits in high-risk zones. "*Some houses are far inside the fields. In the rainy*

*season, we can't go. Even we get scared because of snakes and elephants.*" (PHM, 48) Infrastructure limitations within the health system itself also hampered their effectiveness. Several PHMs highlighted the lack of appropriate medical facilities in their local hospitals. In many cases, bitten children must be transferred long distances to receive definitive care due to the absence of trained staff or antivenom. "*We advise parents to bring their child to the hospital immediately. However, our local hospital does not have facilities for snakebite treatment. They transfer children to the main hospital 40 km away.*" (PHM, 1)

Midwives also noted communication barriers, particularly in linguistically diverse areas. In Tamil-speaking regions, Sinhala-speaking PHMs struggled to engage meaningfully with caregivers due to language mismatches. "*My area is rural and many parents are only Tamil-speaking. Language barriers are there for good communication.*" (PHM, 13) Despite their willingness to educate and guide parents, PHMs reported that the absence of visual or material aids significantly weakened their health promotion efforts. Verbal education alone was often not retained, especially by elderly or less literate community members. "*If we had posters, flip charts, or kits to show them, it would be easier. Now we only talk, and they forget.*" (PHM, 36)

Midwives also encountered structural and policy-level barriers, such as the lack of consistent internet access in rural zones, limited funding for community education initiatives, and what they perceived as insufficient policy attention to snakebites as a public health issue. "*There is a lack of interest about snakebite among policymakers. Even we don't have proper internet connections in our areas to attend training or download materials.*" (PHM, 11)

Additionally, cultural norms and household-level risk factors posed barriers to effective engagement. PHMs described visiting homes where children slept in unsafe conditions, such as unplastered rooms covered with plastic sheeting, which inadvertently attracted or concealed snakes. "*We guide parents to take steps to prevent snakebites in their children. Sometimes their houses are not plastered so they use white polythene to cover baby room. They have found snakes in these areas.*" (PHM, 38)

Nonetheless, midwives expressed a strong sense of commitment and a readiness to contribute more actively—provided they are given appropriate training, educational tools, and structured integration of snakebite topics into their existing service delivery platforms. "*If you give us a plan and materials, we can do it step by step during our routine field visits.*" (PHM, 22) This willingness reflects the untapped potential of Sri Lanka's PHM network to serve as a foundational pillar in community-based snakebite prevention and first aid programs. However, for this potential to be realised, it is essential that their operational constraints, knowledge gaps, and contextual barriers are adequately addressed within health system strengthening initiatives.

**5. Recommendations for system improvement.** The final theme focused on PHMs' perspectives on how to enhance both the prevention and emergency response to paediatric snakebites in their communities. Drawing from their frontline experiences and insights into local dynamics, PHMs offered a wide range of pragmatic and actionable recommendations targeting families, schools, healthcare institutions, and broader policy structures. A consistent and emphatic recommendation was the urgent need to scale up awareness-building efforts at the community level. PHMs advocated for sustained and age-appropriate education targeting children, parents, and teachers. They particularly emphasized the value of integrating snakebite prevention into school-based health education initiatives, both as a means of knowledge transfer and behaviour change. "*If we teach schoolchildren, they will remind their parents and siblings what to do. They learn fast and tell others.*" (PHM, 27)

They also saw untapped potential in using existing community platforms, including religious gatherings, farmer societies, and women's groups to disseminate key prevention and first aid messages. "*People gather in mosques and temples. That is the time to talk to them, not wait for them to come to us.*" (PHM, 62) "*During mother and baby clinics or school health days, even if we speak for five minutes about snakebites, it can make a difference.*" (PHM, 10)

Beyond prevention, midwives identified systemic gaps in emergency response infrastructure. They expressed strong support for the 1990 ambulance service but highlighted critical limitations in its reach and responsiveness—particularly in

remote villages with poor roads, roaming elephants, and a single ambulance covering multiple regions. "*At night, people fear going out because of elephants. The 1990 service is good, but we need more vehicles and faster response.*" (PHM, 8) "*Some families live very far, and the ambulance can't always get there quickly. More local options would help.*" (PHM, 32)

Several PHMs proposed the establishment of specialized treatment units within peripheral hospitals equipped with trained staff and consistent antivenom availability. This would help avoid delays caused by long-distance referrals to tertiary facilities. "*Now we have to send everyone to the main hospital. If the small hospital had antivenom, it would save time.*" (PHM, 22)

Midwives also suggested the creation of better coordination systems during emergencies, including digital tools and real-time communication channels. "*We propose developing an app to facilitate easier collaboration among midwives and other healthcare professionals.*" (PHM, 21) "*If we have an emergency hotline number, we can directly connect with the hospital, inform them about the patient's condition, and they can guide us on the appropriate actions to take in that situation.*" (PHM, 50)

At the policy level, PHMs emphasized the need to embed snakebite prevention and management into routine public health functions. This included increasing the workforce, improving accountability, and formalizing responsibilities for monitoring environmental risk factors like snake breeding sites. "*Increase the number of PHMs and PHIs (Public Health Inspectors) in high-population areas so they can monitor and identify snake breeding sites. If breeding sources are found outside houses, responsible individuals can be fined.*" (PHM, 31) There was also concern about attitudinal barriers among overburdened healthcare staff. PHMs called for a shift in institutional culture to prioritize neglected rural health issues like snakebites. "*Policies are adequate but the mindsets of healthcare professionals need to be changed. Most of them take these issues lightly because they have plenty of other work. If we increase the number of staff we can do a better service.*" (PHM, 23) Training and re-training featured prominently in the recommendations. PHMs emphasized that a one-off awareness session was insufficient to ensure preparedness. Instead, they advocated for continuous professional development, ideally involving hands-on practice, simulations, role-playing, and locally contextualized printed or video materials.

## Discussion

This study provides one of the few qualitative explorations into frontline, field-level experiences of paediatric snakebite prevention and management in Sri Lanka, offering a rare and nuanced view that spans personal beliefs, practical realities, cultural influences, and policy-level barriers. Through in-depth focus group discussions with Public Health Midwives working in two high-incidence districts, we captured a broad tapestry of perspectives that illuminate how snakebite risk, prevention, and care are understood and navigated within rural communities. The findings reveal not only the operational challenges and systemic gaps that limit effective response, but also the embedded cultural beliefs and social dynamics that shape care-seeking behaviour. While PHMs demonstrated a strong sense of responsibility and enthusiasm for engaging in prevention, their experiences underscore the need for contextually grounded, health system–integrated interventions that address the interplay between biomedical guidance, cultural traditions, and structural constraints.

One of the most consistent themes was the limited understanding among PHMs regarding paediatric-specific risk factors and clinical manifestations of snake envenoming. This reflects broader national and regional trends where snakebite-related health education is often generalised, lacking paediatric focus, and not systematically integrated into primary care worker curricula [12–14]. The absence of structured surveillance or notification systems for paediatric snakebites further exacerbates the issue, limiting opportunities for targeted prevention or rapid response. Another prominent barrier identified was the persistence of traditional beliefs and reliance on indigenous healing practices [15,16]. These insights align with prior studies from South Asia and sub-Saharan Africa, which have demonstrated that cultural beliefs and social norms can strongly shape care-seeking behaviours, especially in rural, low-literacy populations [17–19]. The PHMs' call for respectful, locally grounded counter-narratives echoes global recommendations for community-based

interventions in regions with high-risk for snakebites [20]. Despite their commitment, PHMs face major challenges—such as heavy workloads, lack of materials, and poor mobility—highlighting the need for greater institutional support, as seen in similar LMIC settings [21–23].

Notably, the experiences and recommendations of PHMs in this study have implications beyond the Sri Lankan context. In many countries within the global snakebite belt, community health workers occupy a similar frontline role, yet face parallel constraints due to limited training, resource scarcity, and cultural complexities [24]. Addressing these deficits through targeted training, simplified tools, and child-focused educational materials is critical to empowering community health workers and improving outcomes for snakebite-prone children in rural settings. The integration of paediatric snakebite awareness and first aid into existing community health frameworks presents a scalable and cost-effective strategy for other low- and middle-income countries.

## Limitations

This study is not without limitations. First, it was geographically restricted to two high-incidence districts, which may limit generalisability to other parts of Sri Lanka. However, the regions chosen reflect typical rural health system structures and cultural practices found in many snakebite-endemic areas. Second, the study focused exclusively on PHMs and did not capture the perspectives of other stakeholders such as traditional healers. Including such voices in future research could provide a more holistic view of the community-level ecosystem of snakebite response. Lastly, the themes identified largely reflected the domains used to structure the focus group guide and survey; while this consistency supports the salience of these issues, it may also reflect the influence of a pre-determined framework, potentially limiting the emergence of entirely new themes.

## Conclusion

Public Health Midwives are well positioned to support paediatric snakebite prevention in rural Sri Lanka but currently face gaps in knowledge, training, and operational resources. Strengthening PHM capacity through brief, paediatric-focused training and integrating snakebite education into routine maternal–child health activities could substantially improve community preparedness. These findings offer practical, scalable strategies for reducing paediatric snakebite morbidity in similar rural settings across the global snakebite belt.

## Supporting information

**S1 File. Interview guide.**
(DOCX)

**S2 File. COREQ checklist.**
(PDF)

## Acknowledgments

The authors would like to thank the participants for their time and willingness to share their experiences and insights.

## Author contributions

**Conceptualization:** Kavinda Dayasiri.

**Formal analysis:** Kavinda Dayasiri, Tharuka Perera.

**Funding acquisition:** Kavinda Dayasiri.

**Investigation:** Kavinda Dayasiri, Achila Ranasinghe, Tharuka Perera.

**Methodology:** Kavinda Dayasiri.

**Project administration:** Kavinda Dayasiri.

**Supervision:** Kavinda Dayasiri, Indika Gawarammana, Shaluka Jayamanne.

**Writing – original draft:** Kavinda Dayasiri.

**Writing – review & editing:** Kavinda Dayasiri, Achila Ranasinghe, Tharuka Perera, Indika Gawarammana, Shaluka Jayamanne.

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
