## [Decision Letter · Decision Letter 0]

28 Jul 2025

Bridging tradition and evidence: community health workers’ perspectives on paediatric snakebite management in high-risk rural Sri Lankan districts

Dear Dr. Dayasiri,

Thank you for submitting your manuscript to PLOS Neglected Tropical Diseases. After careful consideration, we feel that it has merit but does not fully meet PLOS Neglected Tropical Diseases's publication criteria as it currently stands. Therefore, we invite you to submit a revised version of the manuscript that addresses the points raised during the review process.

Please submit your revised manuscript within 60 days Sep 26 2025 11:59PM. If you will need more time than this to complete your revisions, please reply to this message or contact the journal office at plosntds@plos.org. Please include the following items when submitting your revised manuscript:

We look forward to receiving your revised manuscript.

Kind regards,

Udhishtran Arudchelvam

Guest Editor

Victoria Brookes

Section Editor

Shaden Kamhawi

co-Editor-in-Chief

Paul Brindley

co-Editor-in-Chief

**Additional Editor Comments:**

Dear Corresponding Autor,

Your article “Bridging tradition and evidence: community health workers’ perspectives on paediatric snakebite management in high-risk rural Sri Lankan districts”- is a very important and indeed very neglected area of healthcare, that requires an insight to address the reasons for the delays in seeking healthcare services in countries like Sri Lanka.

Authors are requested to go through the comments of the five reviewers and respond to them and revise the article wherever required and resubmit.

**Journal Requirements:**

At this stage, the following Authors/Authors require contributions: Kavinda Chandimal Dayasiri. Please ensure that the full contributions of each author are acknowledged in the "Add/Edit/Remove Authors" section of our submission form.

3) Some material included in your submission may be copyrighted. According to PLOSu2019s copyright policy, authors who use figures or other material (e.g., graphics, clipart, maps) from another author or copyright holder must demonstrate or obtain permission to publish this material under the Creative Commons Attribution 4.0 International (CC BY 4.0) License used by PLOS journals. Please closely review the details of PLOSu2019s copyright requirements here: PLOS Licenses and Copyright. If you need to request permissions from a copyright holder, you may use PLOS's Copyright Content Permission form.

Potential Copyright Issues:

- Figure 1. Please (a) provide a direct link to the base layer of the map (i.e., the country or region border shape) and ensure this is also included in the figure legend; and (b) provide a link to the terms of use / license information for the base layer image or shapefile. We cannot publish proprietary or copyrighted maps (e.g. Google Maps, Mapquest) and the terms of use for your map base layer must be compatible with our CC BY 4.0 license.

**Reviewers' Comments:**

Reviewer's Responses to Questions

**Key Review Criteria Required for Acceptance?**

**Methods**

-Are the objectives of the study clearly articulated with a clear testable hypothesis stated?

-Is the study design appropriate to address the stated objectives?

-Is the population clearly described and appropriate for the hypothesis being tested?

-Is the sample size sufficient to ensure adequate power to address the hypothesis being tested?

-Were correct statistical analysis used to support conclusions?

-Are there concerns about ethical or regulatory requirements being met?

Reviewer #1: Nil issues

Reviewer #2: The study’s objectives are clearly stated and well aligned with the research focus. Focus group discussions are appropriate for exploring community health workers’ perspectives. The population and data collection are described in detail.

The methodology demonstrates strong rigor, with effective use of reflexivity, triangulation, and member-checking. Analysis is clearly presented, supported by a code tree and evidence of thematic saturation. Ethical clearance is properly obtained; no ethical concerns identified.

This study appears to be part of broader research on pediatric snakebite management in rural Sri Lanka including physician perspectives, parental knowledge, and preventive practices. Briefly stating this context would help situate the current study and highlight its contribution within a comprehensive research effort.

Reviewer #3: The objective of this study was to explore the insights of Public Health Midwives (CHWs) into pediatric snakebite prevention and management in 2 high-incidence districts in Sri Lanka, focus group interviews with qualitative analysis was an appropriate design for this state objective. The authors clearly describe the study participants and with 11 focus groups and a total of 74 participants this is an impressive sample particularly for qualitative analysis. Authors clearly outlined their process for meeting with ethical requirements.

Reviewer #4: Yes

Reviewer #5: Important components are missing in the methodology followed, see questions below

**Results**

-Does the analysis presented match the analysis plan?

-Are the results clearly and completely presented?

-Are the figures (Tables, Images) of sufficient quality for clarity?

Reviewer #1: This is very long and the readership of this journal is going to struggle to read thru, and in fact may only read No 1 and miss much of the important material. This should be shortened (maybe to a 1/3 of the length), and the remaining material needs to be in a Supplement. This is not a qualitative research journal.

Reviewer #2: The results are clearly and well presented, highlighting key barriers such as knowledge gaps, limited experience, and structural barriers. The recommendation for pediatric-specific training and broader system improvements is well justified. A second-order analysis has been conducted, adding depth to the findings.

Figure 1 – This same figure appears to have been previously published in:

Dayasiri, K., Perera, T., Gawarammana, I., & Jayamanne, S. (2025). Caught between fear and tradition: parental knowledge, beliefs and emergency responses to paediatric snakebites in rural Sri Lanka. BMJ Paediatrics Open, 9(1), e003658. https://doi.org/10.1136/bmjpo-2025-003658.

Please clarify whether this is a reused figure and, if so, ensure proper attribution and confirm that permission has been obtained for reuse in the current manuscript.

Reviewer #3: Thematic analysis is an appropriate strategy for this qualitative study; but I do think the authors could be a bit more creative and effective in methods of presentation and visualization of qualitative data (eg thematic maps or network). The results were presented in a coherent and organized fashion and the use of direct quotes was highly effective.

Reviewer #4: Yes, but see the comments

Reviewer #5: No, authors discuss data in the results which combined with the methodology makes it even harder to assume information from FGD was really participant driven. And these discussions are then repeated in discussion section.

**Conclusions**

-Are the conclusions supported by the data presented?

-Are the limitations of analysis clearly described?

-Do the authors discuss how these data can be helpful to advance our understanding of the topic under study?

-Is public health relevance addressed?

Reviewer #1: Good

Reviewer #2: The conclusions are supported by the data presented. However, the section could be strengthened by more explicitly outlining how the suggested improvements—such as pediatric-specific training and system-level changes—might be implemented in practice, as partly discussed in the results and discussion sections.

The limitations are appropriately acknowledged. The authors clearly demonstrate how the findings contribute to advancing our understanding of pediatric snakebite management from the perspective of healthcare workers. The study has clear public health relevance, particularly for improving care in rural, high-risk settings within the global snakebite belt.

Reviewer #3: Discussion and conclusions are well-summarized and consistent with data presented in the study, although I do think they should strengthen the emphasis in their conclusion regarding providing adequate structural supports and addressing the challenges with limited health workforce and competing priorities that were highlighted in the results and are a well-documented challenge with continue to layer on additional programs and responsibilities on frontline polyvalent CHWs. Authors highlighted limitations in terms of only including PHMs in the focus groups. I would love to see the authors strengthen the conclusions with immediate next steps based on these study findings (ie developing and piloting a pediatric snake bite intervention focused on equipping PHMs) given the public health relevance.

Reviewer #4: See comments

Reviewer #5: No, the midwives indicate being overburdened and needed extra staff to be able to pick up snakebite education and implementation. They also suggest other community platforms instead. Yet the researchers state the midwives to be an underutilised workforce and suggest global involvement of this workforce without further context.

**Editorial and Data Presentation Modifications?**

Reviewer #1: Abbreviations: avoid these, particularly things like FGD because qualitative research will not be familiar to many readers and it should be written in full.

Please use the terms snake and snakebite correctly. This is an unfortunately use of English that is creeping into the snake literature. There is no such thing as “Snakebite envenoming” – it should be “snake envenoming” or snakebite that causes envenoming. And the term “snakebite endemic” is even worse. Describe what you mean – endemic means that it comes from that area – of course this is the case with snakebite like this, unless the authors are distinguishing this from exotic snakebite. Any district will have “endemic” snakebites. I assume the authors simply mean that there is a high rate of snakebites in these regions – just say that.

Abstract

The conclusion is long and needs to be shortened with the key points, and not simply restating the Results.

Reviewer #2: (No Response)

Reviewer #3: It would be helpful to add some more concrete epidemiologic statistics to the introduction on the burden (ie incidence, morbidity, mortality) of pediatric snake bites in Sri Lanka or the region.

I would suggest the authors include the range of number of participants/focus group and in table 1 also include native language of the CHWs. I could not view the supplemental coding tree document.

Please mention how many participants/focus group.

Please clarify if the same investigators who facilitated the FGDs were also involved in the coding/thematic analysis.

Reviewer #4: General comments

This study denotes a very important scientific exercise. The topic is very important, urgent and highly neglected. A much-needed exploration with regard to the public health and paediatrics in the context and beyond. As a qualitative exploration, the study reveals valuable and vast knowledge of a broad scope admitting the limitations. In general, the manuscript is well written except for the introduction component. I would suggest the authors to reveal more the subthemes of the analysis as described below. Detail comments follow.

Comment No. LN Comment

01 Abstract The statement, “Public Health Midwives (PHMs) serve as trusted community health workers and are uniquely positioned to support prevention and early response to paediatric snakebites”: It is quite true that the PHMs are trusted community health workers. However, whether they are uniquely positioned to support and prevent paediatric snake bite should be a finding of this study. The PHM’s serve on routine maternal and child health activities and they have set areas to cover within their services. There have been lot of instances where, PHM’s are overloaded with extra work and they follow these with many constraints. So stating that they are uniquely positioned for this need to be carefully sorted within the research.

02 Abstract Results section can be more improved by expressing one major finding under each theme (you may not elaborate the wordy themes here but can mention briefly the six themes) than just listing the major themes. This could be challenging under the word count, but would upgrade the study much more as the audience could have a glimpse of what the study really reflect.

03 Abstract Could you please make the conclusion concise and focused?

04 Author summary LN 62 The abstract mentions five major themes while here authors state six. Please correct.

Introduction

05 Introduction The introduction of this manuscript is very thin. Ideally the introduction should start broad and consist of stating the problem, implications of the problem and describing the problem in the local context, properly stating the research gap and narrowing down to justifying the study. Within this descriptions it would be valuable if the authors define what age range “paediatric” refers to, describe the epidemiology of paediatric snakebite globally and locally and describe the implications including if any, different manifestations from adults, different manangement and different burden reported briefly.

06 Introduction LN84-90 I would suggest the authors to be more transparent with the scope of work of PHMs. The PHMs are the grassroot level maternal and child health care workers serving eligible families in a community. This will mean that they will cater the families with pregnant women and children aged upto five years. Disease surveillance is not a direct duty applied to PHMs but for Public Health Inspectors. So, whether the PHM’s are motivated, willing and capable to work for snakebite prevention and management should be a finding of the study and not the authors opinion at the beginning. Authors can highlight this as a research gap further. Please be precise and appropriate with regard to this paragraph.

07 LN 110-11 Can the authors justify why they selected only PHMs of the public health officers? Why not PHIs? Specifically relating to their active role in school health promotion given the age range of children covered? Is it for feasibility?

08 LN 123-134 Authors can reduce repetition on purposive sampling a little further.

09 LN 133 Will it be more appropriate to mention 5-7 participants per focus group amounting to 11 FGDs with a total of 74 PHMs?

10 LN172 Authors claim “Triangulation of data sources”. The data source in the study is the FGDs of PHMs. Is the above statement valid as there are no multiple data sources in the study?

Results

11 LN 176 It is really good to describe the participants. As this is a qualitative manuscript, I would recommend to briefly include the personal characteristics of participants in a paragraph text rather than a table. But author’s/Editors can decide on this. Howmany PHMs took care of Sinhala and other ethnic communities respectively?

12 LN179 Is it an inductive thematic analysis? Or a deductive according the pre-defined themes? It will be good to mention this.

13 LN 180 -192 The authors state the specific risks for paediatric snake bite. Please transfer this part to the introduction and elaborate it more within the introduction as this is the results section. You can also discuss this in the discussion.

14 LN 224-226 Move to discussion

15 LN 271-273 It is of course a challenge in qualitative research presentations to demarcate between the results and the discussion. The statement included “These findings highlight the complex interplay between tradition, familial hierarchy, and healthcare behaviour, suggesting that any intervention aiming to improve paediatric snakebite outcomes must be culturally sensitive and developed with local context in mind. Simply dismissing these beliefs risks alienating communities”, would be better stated as an opinion of the PHM’s rather than the researcher’s in the results section, if that is the case. Otherwise move to the discussion.

16 LN 389 Elaborate the abbreviation PHI

17 LN 407 onwards (results section) The second order analysis: This might be a good attempt, but adds lot of repetition and jargon in the results section (reflecting doubt of over using language models). I would rather incorporate the statements here within the same results section. One reason is anyway the authors reflect the patterns within the previous sections. If the authors believe that this section should be preserved, mention about this in the methods as well. With the relatively long results section, I would suggest either to incorporate this component while presenting the former results or focus on patterns and provide a concise paragraph for the section avoiding repetition. Did the authors find patterns between the PHMs reflections within different community groups/ cultures or geographies? Would be interesting to highlight this.

18 Results (General) The authors describe the results under the major themes. Although we can clearly identify the subthemes within each major theme, these are not separately stated. The current version of presenting the results section is reader friendly than incorporating specific subthemes within the text. However, if the authors can include a figure to showcase the subthemes reflecting different facets under each major theme, the readers would be able to visualize the findings of the study very directly and easily rather than reading the whole article. One of the aims of thematic analysis is to present this clarity and diversity among the participant opinions to the audience. While having a rich data set, It would be valuable to include a thematic figure representing the subthemes.

19 Results (General) Did the researchers find any direct objections/ barriers or negative attitudes reported by the PHMs on incorporating snakebite prevention to their list of duties? Would be interesting to know.

Discussion

20 First paragraph Ideally the first paragraph of a results section should reflect the significance of the study. Rather than highlighting that PHM’s are ready to work for snakebite prevention, which needs to be interpreted more carefully with a broader study involving the health system, this study is unique as the interviews provide us with a broad tapestry of knowledge ranging from personal beleifs, opinions, practical problems, culture and context to policy level barriers on prevention of paediatric snakebite. I guess this is one of the very few qualitative explorations on field level experiences on the topic. Authors can ofcourse state that PHM’s exert responsibility and enthusiasm on providing services. It would be valuable to reflect this depth and gravity in the introductory paragraph of the discussion.

Conclusions

21 LN500-501 I would like the authors to revise the statement “PHMs in rural Sri Lanka are uniquely positioned to lead community-level paediatric snakebite prevention and response” which as I mentioned earlier should be a conclusion of a much broader exploration. But the authors can conclude their willingness, enthusiasm and perceived responsibility, highlighting the pragmatic gaps and valuable policy implications elaborated by them.

22 Discussion (general) Could the authors add a separate paragraph on country level policy gaps identified by the PHMs? Or organize the facts into one para?

23 LN505 Change “offer a promising path” to “may offer a promising path” as such conclusions cannot be driven directly from exploratory qualitative studies.

Other

24 Title The title phrase, “Bridging tradition and evidence” Does not seem to shed light directly on what the study denotes. It also reflect a too similar pattern with the already available publications on different components of the topic.

25 PPIE Could the authors mention about Patient Public Involvement in this study

Reviewer #5: (No Response)

**Summary and General Comments**

Reviewer #1: Nil further

Reviewer #2: Overall, the study provides valuable insights into community-level pediatric snakebite prevention and response, shedding light on both structural and cultural barriers that influence care in high-risk rural settings. However, since this study appears to be part of broader research on pediatric snakebite management in rural Sri Lanka—including physician perspectives, parental knowledge, and preventive practices—briefly stating this context would strengthen the manuscript.

Specific comments:

Line 230 – The abbreviation “PHM” has already been introduced multiple times (e.g., Lines 28, 58, 85, 111, 187, 299, and 349). Consider avoiding repeated redefinitions unless necessary for clarity.

Line 389 – The abbreviation “PHIs” is unclear. Please clarify its meaning at first use—does it refer to Public Health Inspectors?

Reviewer #3: This is a well-designed and well-written study on the perspectives of CHWs on pediatric snake-bite prevention and management in a high-incidence region of Sri Lanka. While the methods were not particularly novel, it was a very well-conducted and clearly described and I think this community-grounded approach was highly effective in better understanding the nuances of pediatric snake bite perspectives as well as identify potential opportunities for intervention.

Reviewer #4: General comments

This study denotes a very important scientific exercise. The topic is very important, urgent and highly neglected. A much-needed exploration with regard to the public health and paediatrics in the context and beyond. As a qualitative exploration, the study reveals valuable and vast knowledge of a broad scope admitting the limitations. In general, the manuscript is well written except for the introduction component. I would suggest the authors to reveal more the subthemes of the analysis as described below. Detail comments follow

Reviewer #5: Study looking at the role of public health midwives as trusted community health workers and their potential in supporting snakebite prevention and response.

Introduction: the context is missing; on snakebite incidence in children and on health systems in communities; what’s the role of the midwife and how does this compare to other health care providers in the community setting. We don’t know why the authors decided to focus on the midwives.

Methods: Please use a checklist such as COREQ to ensure adequate reporting of this qualitative report. This will help identifying missing information, such as

- What was the methodological orientation stated to underpin the study eg grounded theory, discourse analysis, ethnography, phenomenology, content analysis

- How were participants selected, how many refused to participate etc

Interview: the interview was added but there’s no info on prompts used and only few questions are listed. Without the interview guide and prompts it’s difficult to see how researchers ensured information collected was coming from the participants instead of being influenced/steered in a direction by the interviewers.

There’s no information on data saturation.

Results:

Table 1 please present data as continuous variables instead of in categories, eg mean age with SD instead of grouping.

There’s a lot of overlap in the results and discussion. A substantial part of the results section is actually a discussion (eg 271-276 or 344-348). These then get repeated in the discussion section.

Themes identified are exactly same as topics used to structure the survey. Which demonstrate the limitations of the missing methodological orientation. Why did the researchers not use a more participatory approach and technique in the focus group discussion?

Why do the authors think it’s problematic not being able to identify venomous vs non-venomous bites? Thinking about prevention it may be safe to avoid human-snake encounters with all snakes?

The findings suggest the researchers think midwives should be responsible for reporting paediatric incidents?

We do not know how big the snakebite envenoming problems is compared to the other probably more frequent medical problems the PHMs are working on. How can the researchers justify it’s medical beneficial to take time from midwives dedicated to a less common problem in their communities; which disease/intervention/activity would need to be reduced?

Why do authors believe a intervention looking at prevention and snakebite response would require an effort focused on children only and another one focused on adults? How likely is it for a neglected tropical disease with limited budget to be able to design such interventions for different target groups instead of an overall prevention intervention in the community?

In the abstract the authors conclude the PHMs to represent and ‘underutilized’ workforce. Their findings nevertheless suggest the PHMs to be an ‘overutilized’/ overburdened workforce not able to pick up extra tasks without further investments in the size of the workforce. PHMs indicate to be overburdened and seem to instead suggest to utilize other community platforms to teach communities on prevention and response. There’s a mismatch between the reported statements from the PHMs and the researchers’ conclusions.

Researchers suggest midwives to play a role in similar settings across the global snakebite belt but nowhere in the paper do the researchers discuss other published literature on the role of community health workers in snakebite response or in other NTDs (eg midwife roles in other NTDs or community health volunteers in snakebite response). Without this context it’s difficult to follow the researchers’ conclusion that other high risk snakebite settings should switch their community activities from the community health workers to the midwives.

**Figure resubmission:**

**Reproducibility:**



---

## [Decision Letter · Decision Letter 1]

16 Nov 2025

Response to Reviewers
Revised Manuscript with Track Changes
Manuscript

Shaden Kamhawi

co-Editor-in-Chief

Paul Brindley

co-Editor-in-Chief

**Reviewers' comments:**

**Key Review Criteria Required for Acceptance?**

**Methods**

-Are the objectives of the study clearly articulated with a clear testable hypothesis stated?

-Is the study design appropriate to address the stated objectives?

-Is the population clearly described and appropriate for the hypothesis being tested?

-Is the sample size sufficient to ensure adequate power to address the hypothesis being tested?

-Were correct statistical analysis used to support conclusions?

-Are there concerns about ethical or regulatory requirements being met?

Reviewer #1: The Methods are improved.

Reviewer #2: No issues were identified; all suggested changes have been implemented.

Reviewer #3: Methods were clearly explained and authors addressed feedback provided in previous review.

Reviewer #4: yes

**Results**

-Does the analysis presented match the analysis plan?

-Are the results clearly and completely presented?

-Are the figures (Tables, Images) of sufficient quality for clarity?

Reviewer #1: The Results are very long, and almost no reader of this journal will read them all. Please reduce this and use supplementary material to provide the information. In fact the Results could be much shorter, possibly even the first couple of paragraphs with reference to supplementary material. This is not a qualitative research journal.

Reviewer #2: No issues were identified; all suggested changes have been implemented.

Reviewer #3: Addition of thematic map strengthened the clarity of the analysis.

Reviewer #4: The supplementary figure on the thematic map added to the manuscript is valuable for the direct visualization of the study's findings. However, the quality of the figure is substandard. I would recommend that the authors prepare a better figure using the freely available softwares and include this figure not as supplementary but as a main figure in the manuscript.

**Conclusions**

-Are the conclusions supported by the data presented?

-Are the limitations of analysis clearly described?

-Do the authors discuss how these data can be helpful to advance our understanding of the topic under study?

-Is public health relevance addressed?

Reviewer #1: The conclusions are far too long, two long paragraphs. You need to summarise the study into 3-4 sentences, so there are take home easy to understand messages. It also needs to be clinically and epidemiologically applicable.

Reviewer #2: No issues were identified; all suggested changes have been implemented.

Reviewer #3: With the revisions authors more clearly supported conclusions with data and highlighted more clearly both the potential and the structural barriers to deploy PHMs (a cadre of CHWs) in addressing pediatric snakebites.

Reviewer #4: Yes

**Editorial and Data Presentation Modifications?**

Reviewer #1: The authors need to have the manuscript read by a native English speaker to improve the grammar.

The introduction still contains the term "snakebite envenoming". This should be either "snakebite", which is when a human is bitten by a snake OR "snake envenoming" in which a snakebite results in envenoming. Please correct this throughout as originally asked.

The author continue to use the term "endemic", this is not an appropriate term for snakebite in a region such as Asia. There are snakes everywhere, and this should simply say in regions in which the incidence of snakebite is high. This is a very important distinction, because "endemic" means nothing in this context, and high rates of snakebite are due to a high rate of the interaction between snakes and humans - which has nothing to do with whether the snakes are endemic or not. Again, this is a misuse of the English and the authors needs to get advice from an English speaker with expertise in snakebite terminology.

The authors state that paediatric snake bite ... "with higher risks of severe envenoming, long-term disability, and death compared to adult". I am unaware of any evidence to support this. The reference 3. Abouyanis et al. is a study of only paediatric intensive care patients, and no adult snakebite patients. It simply says that the burden of paediatric snakebite is high in relation to other paediatric conditions. They do not compare it to adults and in fact say that the mortality was low. So please remove this because it is untrue, and not a reason to justify a study in paediatrics.

Reviewer #2: No issues were identified.

Reviewer #3: (No Response)

Reviewer #4: (No Response)

**Summary and General Comments**

Reviewer #1: (No Response)

Reviewer #2: Following revision, the manuscript demonstrates improved quality and is recommended for publication.

Reviewer #3: Pediatric snakebites are an important public health problem and the authors highlighted the importance of a community-based approach to addressing gaps in education and early interventions for snakebites with the revisions the authors more clearly stated how this fits into a larger ongoing program to strengthen pediatric snakebite prevention and management. This qualitative study with PHMs on pediatric snakebite prevention and management was well-designed and adds valuable contribution to the field.

Reviewer #4: This manuscript reflects a commendable effort in addressing a significant public health issue. The authors have used and reported standard methods, which are exemplary to the research community conducting qualitative health research on NTDs. The study generates vast local knowledge that can be readily applied to disease prevention and control. I recommend this study for publication with the above suggested minor revision of the supplementary figure on the thematic map.

PLOS authors have the option to publish the peer review history of their article (what does this mean? ). If published, this will include your full peer review and any attached files.

**Do you want your identity to be public for this peer review?** For information about this choice, including consent withdrawal, please see our Privacy Policy .

Reviewer #1: No

Reviewer #2: No

Reviewer #3: No

Reviewer #4: **Yes:** Thilini Agampodi

**Figure resubmission:**

**Reproducibility:** To enhance the reproducibility of your results, we recommend that authors of applicable studies deposit laboratory protocols in protocols.io, where a protocol can be assigned its own identifier (DOI) such that it can be cited independently in the future. Additionally, PLOS ONE offers an option to publish peer-reviewed clinical study protocols. Read more information on sharing protocols at https://plos.org/protocols?utm_medium=editorial-email&utm_source=authorletters&utm_campaign=protocols

---

## [Editor Report · Decision Letter 2]

27 Jan 2026

Dear Dr. Dayasiri,

We are pleased to inform you that your manuscript 'Public Health Midwives’ perspectives on paediatric snakebite prevention and management in rural Sri Lanka: a qualitative study' has been provisionally accepted for publication in PLOS Neglected Tropical Diseases.

Best regards,

Abhay R Satoskar

Section Editor

Victoria Brookes

Section Editor

Shaden Kamhawi

co-Editor-in-Chief

Paul Brindley

co-Editor-in-Chief

---

## [Editor Report · Acceptance letter]

Dear Dr. Dayasiri,

We are delighted to inform you that your manuscript, "Public Health Midwives’ perspectives on paediatric snakebite prevention and management in rural Sri Lanka: a qualitative study," has been formally accepted for publication in PLOS Neglected Tropical Diseases.

Best regards,

Shaden Kamhawi

co-Editor-in-Chief

Paul Brindley

co-Editor-in-Chief
